# Allele Frequency of the C.5G>A Mutation in the *PRCD* Gene Responsible for Progressive Retinal Atrophy in English Cocker Spaniel Dogs

**DOI:** 10.3390/ani9100844

**Published:** 2019-10-21

**Authors:** Larissa R. Andrade, Amanda M. Caceres, Anelize S. Trecenti, Claudia Valeria S. Brandão, Micaella G. Gandolfi, Evian V. Aguiar, Danilo G.A. Andrade, Alexandre S. Borges, Jose P. Oliveira-Filho

**Affiliations:** 1Department of Veterinary Clinical Science, School of Veterinary Medicine and Animal Science, São Paulo State University (Unesp), Botucatu 18618-681, Brazil; andradelrm@gmail.com (L.R.A.); amandamanarac@gmail.com (A.M.C.); anelize.ast@gmail.com (A.S.T.); valeria.brandao@unesp.br (C.V.S.B.); mi_gandolfi@hotmail.com (M.G.G.); alexandre.s.borges@unesp.br (A.S.B.); 2São Paulo State University (Unesp), Medical School, Botucatu 18618-687, Brazil; eva_aguiar@hotmail.com

**Keywords:** diagnosis, genetic disease, genotyping, prcd-PRA

## Abstract

**Simple Summary:**

Progressive retinal atrophy (PRA) in English cocker spaniels (ECSs) is associated with progressive rod–cone degeneration (prcd-PRA), an inherited autosomal recessive disease caused by the c.5G>A mutation in the progressive rod–cone degeneration (*PRCD*) gene. Data regarding the prevalence of the mutated allele are scarce in the global literature, and there is no study evaluating this frequency in Brazil. Therefore, the aim of this study was to evaluate the allele frequency of the c.5G>A mutation in the *PRCD* gene responsible for progressive retinal atrophy (prcd-PRA) in ECS dogs.

**Abstract:**

Progressive retinal atrophy (PRA) due to the c.5G>A mutation in the progressive rod–cone degeneration (*PRCD*) gene is an important genetic disease in English cocker spaniel (ECS) dogs. Because the prevalence of this disease has not been verified in Brazil, this study aimed to evaluate the allele frequency of the c.5G>A mutation in the *PRCD* gene. Purified DNA from 220 ECS dogs was used for genotyping, of which 131 were registered from 18 different kennels and 89 were unregistered. A clinical eye examination was performed in 28 of the genotyped animals; 10 were homozygous mutants. DNA fragments containing the mutation region were amplified by PCR and subjected to direct genomic sequencing. The prcd-PRA allele frequency was 25.5%. Among the registered dogs, the allele frequency was 14.9%; among the dogs with no history of registration, the allele frequency was 41%. Visual impairment was observed in 80% (8/10) of the homozygous mutant animals that underwent clinical eye examination. The high mutation frequency found in this study emphasizes the importance of genotyping ECSs as an early diagnostic test, especially as part of an informed breeding program, to avoid clinical cases of PRA.

## 1. Introduction

Retinal diseases are among the most frequent and best characterized inherited ocular disease in in dogs [1]. The term progressive retinal atrophy (PRA) refers to the group of genetic diseases found in more than 100 dog breeds, and approximately 20 mutations in 18 different genes are known to be responsible for PRA [2,3].

The onset of clinical signs of the disease and its progression vary among affected breeds [2]. Clinical diagnosis is observed in English cocker spaniel (ECS) dogs between 3 and 13 years of age [4]. The disease is clinically characterized by the loss of night vision (nictalopia), which is caused by rod cell degeneration, progressively evolving to the loss of day vision (hemeralopia) as a consequence of cone degeneration, eventually leading to bilateral total blindness [5,6,7]. Alterations such as hyperreflexia, pigmentation and vascular changes, and optic nerve atrophy are common findings in bilateral fundus examination in the early phase of the disease [2].

Progressive rod–cone degeneration (prcd-PRA), the genetic form of PRA that affects the ECS and other breeds (e.g., Miniature poodle, American cocker spaniel, Portuguese water dogs, and Labrador retriever) is an inherited autosomal recessive disease caused by the c.5G>A mutation in the progressive rod–cone degeneration (*PRCD*) gene [4]. This mutation leads to the substitution of a guanine for an adenine (TGC>TAC), which results in the substitution of a cysteine for a tyrosine in the protein [4,8]. The same genetic mutation responsible for prcd-PRA in dogs causes retinitis pigmentosa in humans, with phenotypic similarities that include retinal vessel attenuation and secondary cataracts. Therefore, dogs with this mutation may serve as an experimental model for the study of the disease in humans [4].

Data regarding the allele frequency of this mutation are scarce in the global literature. In the Czech Republic, an allele frequency of 34% was observed among 135 ECS dogs assessed in 2011 [9]. In Brazil, there is no preview study evaluating this frequency. Although there is a study reporting clinical cases of PRA in ECS dogs, it does not discuss the genotype of these animals [10].

Clinical diagnosis of the disease can be performed by fundoscopy and electroretinography (ERG) [11]. However, because PRA is a late-onset disease, the possibility of identifying dogs with heterozygous and homozygous mutations at an early age based on molecular techniques is an important tool for an informed breeding program that aims to reduce disease prevalence in future generations of dogs [11]. The aim of this study was to evaluate the allele frequency of the c.5G>A mutation in the *PRCD* gene responsible for progressive retinal atrophy (prcd-PRA) in ECS dogs.

## 2. Materials and Methods

This study was performed in accordance with the policies of the School of Veterinary Medicine and Animal Science, Animal Care and Use Committee (0218/2016-CEUA/UNESP), and samples were collected under a strict confidentiality agreement to ensure the anonymity of establishments, owners, and animals.

Because the allele frequency of the prcd-PRA mutation in ECS is unknown in Brazil, we used the allele frequency (12%) previously reported by the English Cocker Spaniel Club of America (ECSCA) in 2017 (ecscahealthandrescue.org/2013-07-04-11-01-05/prcd-pra), which involved a population of 1434 ECSs (as there is no information about the ECS population in Brazil, we used the registrations in the Confederação Brasileira de Cinofilia (CBKC) from 2013 to 2017), a 5% margin of error, and a 95% confidence interval to calculate the sample size using OpenEpi software [12].

Blood or oral swab samples were collected from 220 ECS dogs (151 females and 69 males, with ages ranging from 4 months to 17 years) from four different regions of Brazil. Of these samples, 131 were from dogs registered in the CBKC, and 89 were unregistered. The samples from the registered dogs were collected in kennels and at dog shows; these dogs were from 18 CBKC kennels. The samples from the 89 unregistered dogs were collected at breeding events (30) or were obtained from the DNA database of the Laboratory of Molecular Biology of the Veterinary Clinic from School of Veterinary Medicine and Animal Science/UNESP of Botucatu-SP (59), which includes dogs examined in the Veterinary Hospital of the same institution. During the sampling of the unregistered dogs, the veterinarians checked whether the dogs phenotypically adhered to the breed standard, and only those with the breed’s phenotype were included in the study.

The owners were questioned about any complaint of ophthalmic disease, and the answer was negative for 192 of the dogs. In contrast, 28 dogs (6 registered and 22 unregistered) presenting clinical signs of ophthalmic diseases underwent ophthalmic evaluation. The main question in the anamnesis concerned visual impairment, and the owners were asked if diminished visual acuity was observed (e.g., if their dogs bump into obstacles). The clinical eye examination included menace response, pupil reflexes, and fundoscopy using ClearView^™^ Optical Imaging System (Optibrand^®^, Ft. Collins, CO, USA). Fundoscopy was performed only on animals with corneal and lens transparency. Additionally, ERG was performed with the BPM-200 Retinographics system (Retinographics Inc., Norwalk, CT, USA) on one animal identified as a homozygous mutant.

Genomic DNA was purified from the blood or oral swab samples using a commercial kit (GE Healthcare). Specific primers [4] (forward CCAGTGGCAGCAGGAACC; reverse CCGACCTGCTGCCCACGACTG) were designed to genotype the PRCD_c.5G>A mutation. Polymerase chain reaction (PCR) (25 μL) was performed using a reaction containing 12.5 μL of GoTaq^®^ Green master mix (Promega^©^, Madison, WI, USA), 0.75 μL of each primer, 2.5 μL of template DNA, and 8.5 μL nuclease-free water. The amplification conditions were as follows: initial denaturation at 95 °C for 5 min; followed by 37 cycles of denaturation at 95 °C for 45 s, annealing at 62°C for 35 s, and extension at 72 °C for 35 s; a final extension at 72 °C for 5 min. Amplicons (512 bp) were analyzed by 1.5% agarose gel electrophoresis, purified, and subjected to direct sequencing. The sequences and electropherograms were analyzed using Geneious^®^ software (Biomatters^©^, Auckland, New Zealand).

Allele frequencies were compared between males and females and registered and unregistered dogs, and clinical ophthalmologic signs were compared between the dogs’ genotypes (homozygous mutant versus heterozygous plus homozygous wildtype) by a two-sided Fisher’s exact test using GraphPad Prism software (GraphPad Software^©^, San Diego, CA, USA).

In addition, pedigree visualization and analysis, not only of the registered dogs included in this study but also of three generations of each dog, were performed using Pedigraph v2.4 software [13]. Genealogical information of the dogs was obtained from a pedigree database (https://cockerspanieldatabase.info/en/).

## 3. Results

Of the 220 ECSs assessed, 59% (130/220) were homozygous wildtype (G/G) animals; 31% (68/220) were heterozygous (G/A), and 10% (22/220) were homozygous (A/A) mutant animals (Figure 1). Therefore, the allele frequency of the mutation was 25.5% (Table 1). There was no significant difference between males and females (*p* = 0.5295).

Evaluation of the genotype of the 131 dogs registered in the CBKC showed a mutant allele frequency of 14.9%. In the group of unregistered dogs, the allele frequency of the mutation was 41%, which was significantly higher than in the registered group (*p* < 0.0001). Pedigree analyses, performed only on the group of registered animals, revealed an average inbreeding coefficient of 0.00941778, indicating low consanguinity among them. In addition, in the group of animals without complaints of ophthalmic disease, the mutant allele frequency (22%) was significantly lower than that in the group of dogs with clinical signs of ophthalmic disease (48%) (*p* < 0.0001).

The results of the ophthalmic evaluation performed in 28 animals with clinical signs of ophthalmic disease and an average age of 9.6 years (ranging from 3 to 15 years) are presented in Table 2. The presence of visual impairment (*p* = 0.0033) and cataract (*p* = 0.0344) in affected dogs (A/A) was significantly higher than that in nonaffected dogs (G/G + G/A). Visualization of the fundus was not possible in 13 animals, which did not show corneal or lens transparency due to the presence of cataracts and/or alterations such as corneal opacity. The following fundus alterations were observed in six animals: vessel attenuation, hyperreflexia, hyporeflexia, and pallor of the optic disc. Nine animals had an unaltered fundus (Figure 2). ERG was performed in an eight-year-old affected animal but revealed no rod response in either eye, no cone response in the right eye, and decreased cone response in the left eye.

## 4. Discussion

According to the ECSCA, which compiles data from a molecular diagnostic laboratory, the global allele frequencies of the c.5G>A mutation in 1999 and 2017 were approximately 44% and 12%, respectively (ecscahealthandrescue.org/2013-07-04-11-01-05/prcd-pra). Thus, the reduction in allele frequency has occurred more dramatically since 2012, falling from approximately 25% to 12% in 2017. This decrease is associated with the increased use of genetic tests by breeders and consequent mating selection based on the genotype of the animals.

The allele frequency for this mutation in the UK in January 2019 was 15.1%, according to genotyping of 3403 ECS dogs recorded in The Kennel Club (https://www.thekennelclub.org.uk/health/for-breeders/dna-screening-schemes-and-results/dna-screening-for-breeds-sz/spaniel-cocker-dna-screening/). Although there was a lack of peer review evaluating the allele frequency of prcd-PRA in ECS dogs, the allele frequency of prcd-PRA in this breed was 34% in the Czech Republic in 2011 [9], which is similar to that found in the same year by ECSCA.

In the present study, the overall allele frequency was 25.5%; however, this frequency decreased to 14.9% in the group of dogs registered in the CBKC, whereas the allele frequency increased to 41% in the group of unregistered dogs. Although there are no previous studies in Brazil that compare the allele frequency of previous years, the results of the present study in the group of animals registered in the CBKC resemble those found in ECSCA and The Kennel Club (UK). In contrast to these associations, the CBKC does not provide a list of dogs and their respective genotypes, though this may help in mating efforts to reduce the frequency of the mutation. Nevertheless, the allele frequency observed in the dogs registered in the CBKC was similar to that found in the two associations mentioned above, which may reflect knowledge of the disease and the selection of mating partners by affiliated breeders according to genotype. In contrast, the higher allele frequency found in the group of unregistered dogs, which was significantly different (*p* < 0.0001) when compared to that of the registered group, suggests that nonaffiliated breeders in Brazil are unaware of the importance of the disease in the ECS breed and mate dogs indiscriminately, without knowledge of genotype.

The prcd-PRA mutation is an important cause of blindness and one of the ophthalmic diseases for which the ECS breed has a high predisposition [6]. This was observed in the current study because the allele frequency of the mutation was 48% among the dogs that showed signs of ophthalmic diseases, whereas this frequency was 22% among the dogs that did not present such signs. The other ophthalmic diseases found in homozygous wildtype (n = 10) and heterozygous (n = 7) animals with prcd-PRA corroborate other studies that considered ECS a breed that commonly develops other ocular diseases in addition to prcd-PRA [6].

Visual impairment occurred in 8 of the 10 dogs that were homozygous mutant for prcd-PRA and underwent ophthalmic evaluation; according to the owners, the impairment was observed between 5 and 14 years of age. The pupillary light reflex in animals with prcd-PRA slowly decreases as the disease progresses and may still be present in animals in advanced stages of the disease [3]. In our study, this clinical sign was decreased or absent in four homozygous mutant dogs. Cataracts are a common disorder secondary to PRA and can be diagnosed in ECS dogs in the early stages of PRA [14]. In a previous study, 90% of ECS dogs with cataracts and a decrease in visual acuity also had PRA [10]. Corroborating these authors [10], cataracts were observed in a majority (7/10) of the animals with prcd-PRA in our study who underwent ophthalmic evaluation, preventing fundoscopy examination in four.

The main alterations found by fundoscopy of the four homozygous mutant dogs assessed (i.e., retinal vessel attenuation (4/4), hyperreflexia (2/4), hyporeflexia (2/4), and pallor of the optic disc (2/4)) have been described in the initial course of the disease (retinal vessel attenuation and hyperreflexia) or observed as the disease progressed (hyporeflexia and pallor of the optic disc) [2,3,10]. These alterations, which are suggestive of prcd-PRA, were visualized in the present study in animals ranging from 8 to 14 years, in accordance with other authors [4]. As found in two homozygous mutant dogs in our study, fundoscopy may reveal no alterations in affected animals in the early stages of the disease [3,10,15]. In addition, retinal changes, as shown by the fundoscopy of a homozygous wildtype dog and another heterozygous dog, may be due to other diseases, for example, inflammatory or infectious events, toxin exposure, and vascular diseases [3].

Conditions such as cataracts, keratoconjunctivitis sicca, uveitis, and glaucoma prevented eye fundus visualization in approximately half of the animals, regardless of genotype, that underwent ophthalmologic evaluation (13/27). As in these cases, the impossibility of corneal evaluation reinforces the importance of other complementary tests in addition to fundoscopy for the clinical diagnosis of PRA.

ERG, which evaluates retinal function, was performed for only one homozygous mutant animal at age 8 and revealed no rod response in either eye, no cone response in the right eye, and decreased cone response in the left eye. Moreover, fundoscopy for this animal showed only attenuation of retinal vessels. Therefore, the ERG confirmed the clinical disease in this animal, emphasizing its use as a diagnostic tool that is more sensitive than fundoscopy [3,10,15] because alterations may be found even in the early stages of the disease. Considering that the cost of this examination may render it inaccessible to most owners, a lack of general ERG examination was a limitation of our study. Moreover, the need to subject the dog to sedation and/or general anesthesia for the examination may represent a risk for older dogs and not be authorized by the owners. These conditions prevented the use of this test on the other dogs that underwent ophthalmic evaluation. Nonetheless, ophthalmic exams and genotyping findings showed interesting associations, as visual impairment was observed in 80% (8/10) of the homozygous mutant animals that underwent clinical eye examination.

## 5. Conclusions

Because PRA is a late-onset disease and, in general, has a slow progression, few animals undergo ophthalmic evaluation until they present the first clinical signs. However, as the etiological diagnosis of prcd-PRA is not always achieved by ophthalmic evaluation, genotyping is an indicated and necessary tool. This study was able to establish the current situation of this genetic disease in ECS dogs in Brazil; in view of the scarcity of data regarding its prevalence in the global literature, our findings demonstrate a high allele frequency of the mutation associated with the disease, emphasizing the importance of genotyping as a method of early diagnosis, not only to improve the quality of life of affected dogs and their owners, but also for mating practices aimed at reducing clinical cases of the disease.

## Figures and Tables

**Figure 1 animals-09-00844-f001:**
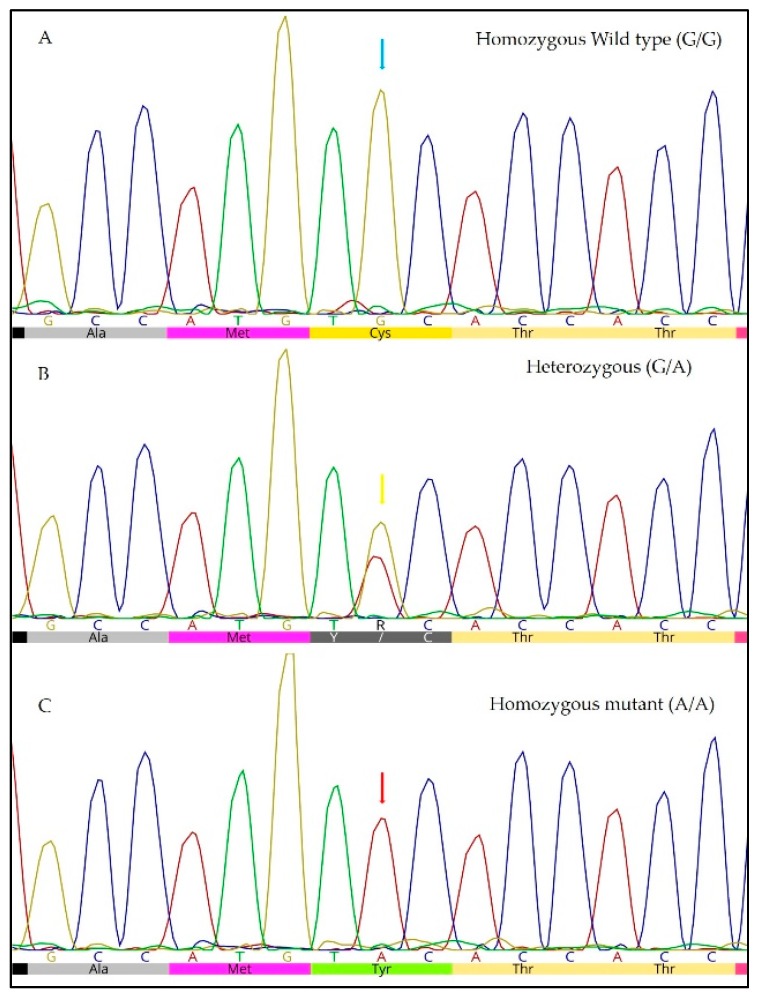
Partial chromatogram showing capillary sequencing results for homozygous wildtype (**A**) and heterozygous (**B**) and homozygous (**C**) mutant alleles of the c.5G>A mutation in the *PRCD* gene in English cocker spaniel dogs. A: Wildtype allele (guanine) (green arrow) and the respective amino acid cysteine (Cys); B: double peak (guanine / adenine, R) is observed (yellow arrow); in addition, note the amino acid cysteine (C) or tyrosine (Y); C: recessive allele (adenine) (red arrow) and the respective changed amino acid tyrosine (Tyr). Image obtained using Geneious^®^ 10.0 software (Biomatters Ltd., Auckland, New Zealand)

**Figure 2 animals-09-00844-f002:**
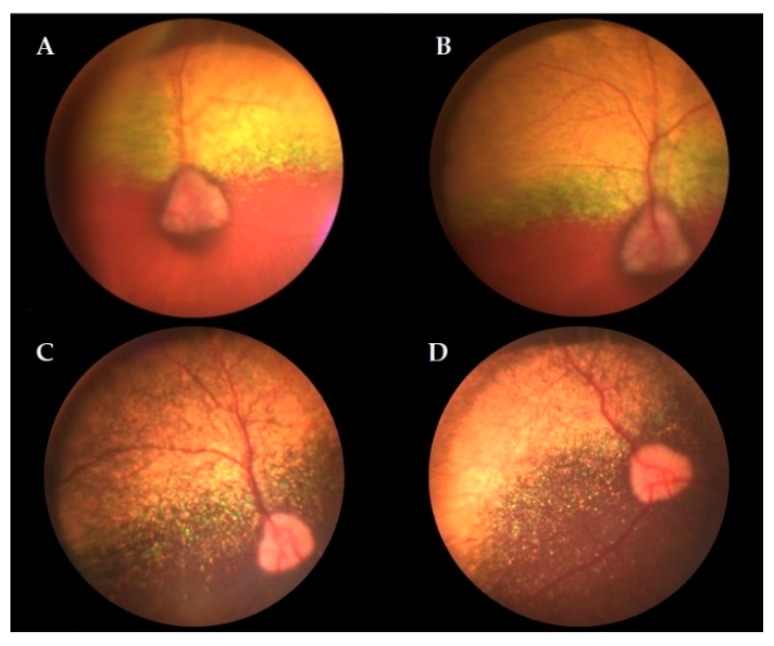
Fundoscopy images of two English cocker spaniel dogs. A and B: Retinal vessel attenuation in the left eye (**A**) and right eye (**B**) of a homozygous mutant dog. C and D: Unaltered fundus in the left eye (**C**) and right eye (**D**) of a homozygous wildtype dog.

**Table 1 animals-09-00844-t001:** Genotyping results for the c.5G>A mutation in the *PRCD* gene responsible for progressive retinal atrophy in English cocker spaniel dogs.

	Homozygous Wildtype (G/G)	Heterozygous (G/A)	Homozygous Mutant (A/A)	Allele Frequency
General analysis	59% (130/220)	31% (68/220)	10% (22/220)	25.5%
Registered dogs	72.5% (95/131)	25.2% (33/131)	2.3% (3/131)	14.9%
Unregistered dogs	39.3% (35/89)	21.4% (19/89)	39.3% (35/89)	41.0%
Without complaint of ophthalmic disease	62% (119/192)	32% (61/192)	6% (12/192)	22.0%
With clinical signs of ophthalmic diseases	39% (11/28)	25% (7/28)	36% (10/28)	48%

**Table 2 animals-09-00844-t002:** Clinical ophthalmologic signs of English cocker spaniel dogs (n = 28) according to the c.5G>A mutation in the *PRCD* gene responsible for progressive retinal atrophy (prcd-PRA).

		Homozygous Wildtype (G/G)	Heterozygous (G/A)	Homozygous Mutant (A/A)	*p*-Value
		(n = 11)	(n = 7)	(n = 10)	(G/G+G/A vs. A/A)
Visual impairment	Present	2 (18%)	1 (14%)	8 (80%)	0.0033
	Absent	9 (82%)	6 (86%)	2 (20%)	
Pupillary light reflex	Present	7 (64%)	4 (57%)	6 (60%)	0.6668
	Decreased	0	2 (29%)	2 (20%)	
	Absent	1 (9%)	1 (14%)	2 (20%)	
	Not visualized *	3 (27%)	0	0	
Cataract	Present	0	3 (43%)	7 (70%)	0.0344
	Absent	8 (73%)	4 (57%)	3 (30%)	
	Not visualized *	3 (27%)	0	0	
Fundoscopy	Altered	1 (9%)	1 (14%)	4 (40%)	0.1357
	Unaltered	6 (55%)	1 (14%)	2 (20%)	
	Not visualized *	4 (36%)	5 (71%)	4 (40%)	

* Not visualized due to pigmentation or corneal opacity or cataract

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
