# Peer review of "Allele Frequency of the C.5G>A Mutation in the PRCD Gene Responsible for Progressive Retinal Atrophy in English Cocker Spaniel Dogs"

_animals, 2019, doi:10.3390/ani9100844_

Round 1

Reviewer 1 Report

Your manuscript describes a straight forward study to estimate the frequency of the c.5G>A mutation in the PRCD gene that is associated with progressive rod cone degeneration (prcd), a form of progressive retinal atrophy (PRA) in English cocker spaniel dogs (ECS). As you correctly point out, very few studies have been published that describe the frequency of this mutation, that is associated with PRA in well over 20 breeds of dog globally, and so is a significant cause of blindness in dogs worldwide. As such I think this study should be of significant interest to a small audience, namely the Brazilian Kennel Club, veterinarians, veterinary ophthalmologists and ECS breeders .

However it would benefit from some changes.

Major modifications:

First, more clarification is required regarding how the dogs were recruited. Estimates of allele frequency rely on the cohort of individuals studied being representative of the population as a whole. There is no indication that these dogs were selected to represent the whole ECS population in Brazil. For example, were the dogs unrelated to one another at any level (parent, grandparent?). Or were they selected from different regions of the country. If not, then the authors need to acknowledge that this cohort might not be representative of the whole breed population. It would be beneficial to state the numerical size of the breed in Brazil.

Second, a lot more clarification is required surrounding the assessment for visual impairment of the dogs studied. Were the signs of visual impairment in 28 dogs owner reported? Similarly, was the absence of signs of visual impairment in the remaining dogs also owner reported? And were all the dogs whose owns reported visual impairment examined by an ophthalmologist? If the owners’ were asked about their dogs’ vision than please include details of the questions they were asked in the manuscript. And what were the qualifications of the individuals who performed the eye examinations? Table 1 is described in the text (line 127) as ‘The results of the ophthalmic evaluation performed in 28 animals with clinical signs of ophthalmic disease…’. But the table indicated that 17 dogs did not have visual impairment. Does that mean that the owners’ assessment was incorrect? Please clarify all these questions in more detail.

Minor modifications:

Throughout the term ‘recessive homozygous’ is used. This is not conventional terminology. It would be better to start by defining homozygous wildtype, heterozygous, and homozygous mutant by their genotypes (i.e. G/G, G/A and A/A) and then refer to G/G (or wildtype), G/A (or heterozygous) and A/A individuals thereafter. Or something equivalent.

Line 32. Replace ‘the high number of mutations’ with ‘The high mutation frequency’.

Line 33. The term ‘breeding orientation’ is not conventional. Please replace with ‘as part of an informed breeding programme’ or something similar.

Line 37. Delete ‘in the group of’, so the sentence reads: Retinal diseases in dogs are among the most frequent and best characterised inherited ocular disease in the species.

Line 39. Replace set with group.

Line 48. Rephrase the first sentence. PRA is not associated with prcd; prcd is a specific genetic form of PRA caused by a specific mutation.

Line 52. Replace exchange with substitution

Line 56. It is incorrect to refer to the prevalence of a mutation. Prevalence refers to a disease. Use frequency to describe a mutation.

Line 60 and onwards. This paragraph needs rephrasing. The main point is that detection of the mutation, using molecule techniques, enables the identification of heterozygous and homozygous dogs at an earlier age than clinical examination can, so is an important tool in the reduction of disease prevalence in future generations of dogs. Heterozygosis and homozygosis are not correct terms to use.

Line 71. Change first sentence to Blood or oral swabs were collected from 220 ECS

Line 82. Ocular ‘means’ is not correct terminology.

Line 87. Synthetized is not a word.

Line 95. The sentence starting ‘The results of the genotype analysis…’ should be deleted. This is not necessary for the Methods section.

Results. Use nomenclature for the different genotypes as defined earlier, and these results would be easier to decipher if summarised in a Table rather than as text.

Discussion.

Line 145. Please define the abbreviation ECSCA.

Line 155. Refer to a lack of peer-review rather than publication

Line 189 onwards. It is inappropriate to refer to the clinical signs associated with disease progression because these dogs were all examined at a single point in time.

The manuscript would benefit from rigorous proof reading by a fluent English speaker.

Author Response

Dear Reviewer 1.

Sincerely,

Reviewer 2 Report

Andrade et al. investigates allele frequency of c.5G>A mutation in PRCD gene from 220 English cocker spaniel dogs in Brazil, and its association with clinical features of progressive retinal atrophy (PRA).

Suggestions:

L53-55: Additional information on the clinical similarities and differences between retinitis pigmentosa in humans and PRA in dogs would be informative. 

L57: Sample size of the dogs used in the 2011 study in Czech Republic will be informative. 

L98,105,121: Include more details on the statistical tests performed, e.g. 1-sided or 2-sided?

Table 1: This table includes interesting observations. Performing appropriate statistical tests and including their results will improve the study’s value substantially. For example, there are clear differences in patterns between homozygotes and wild types / heterozygotes in visual impairment, cataract, and fundoscopy, whereas pupillary light reflex does not seem to present noticeable difference. Appropriate statistical results to see if these differences are meaningful or not will be interesting.

L145: Including the sample size from which the allele frequency was calculated will be informative. 

L167-181: Tabulating the descriptive information on allele frequency across registered, unregistered, healthy and disease groups into a table will help to understand the data better. 

L174-175: Is the allele frequency difference statistically significant? 

L193-197: It is unclear what the key message is.

L198-203: Please break it down to multiple sentences for readability. 

L212-216: Please break down to multiple sentences for readability.

Author Response

Dear Reviewer 1.

Sincerely,

Reviewer 3 Report

The authors of the manuscript submission entitled “Allelic frequency of the c.5G>A mutation in the PRCD gene responsible for progressive retinal atrophy in English cocker spaniel dogs” present observational data for allele frequencies on a small number of English cocker spaniel (ECS) dogs in Brazil.  There are a number of substantial, fatal flaws with the work not the least of which is the small number of dogs they genotyped for the c.5G>A mutation. The number assessed (220) is insufficient to draw population conclusions especially as the authors’ sample included ~40% that were unregistered and may not have been purebred—the lack of congruency in the allele frequencies indicate separate populations which further weakens to the power of their sampling.  Nor did the authors provide any indication as to the actual population size of the ECS in Brazil (registered or dogs that appear to phenotypically resemble ECS).  Furthermore, the authors fail to provide compelling rationale as to the reason to publish allele frequencies of a single gene; stating it has not been done for Brazil is not an adequate reason.  The finding of only 28 dogs with some sort of clinical ophthalmic sign and then not fully linking it to progressive retinal atrophy caused by the mutation is confusing.  The authors also indicate that their ophthalmic exams could not diagnose properly. Are the authors implying that the c.5G>A mutation genotype is not causal for PRA in the ECS?  Then they say it is important to use the genetic test prior to symptoms being expressed—which is why the test exists in the first place. 

Author Response

Dear Reviewer 3.

Sincerely,

Round 2

Reviewer 3 Report

The authors have not adequately addressed the reviewers concerns.  Additionally, the Title is misleading as the frequencies were only done in Brazilian (and a tiny subset at that) English Cocker Spaniels so the region and populations evaluated must be included. Importantly, the authors still have not addressed the utility of their findings in any substantive way in the revised manuscript. This manuscript offers no unique investigative insight beyond what has been published previously.

The authors claim in their response to reviewer one that the samples are representative of the nation and yet in the response to reviewer 3 that only a few Brazilian states are represented and the majority of the samples are from two states. The authors still fail to detail how many dogs of this breed are even registered in the country to give an idea of the proportion of the overall population the 132 registered dogs represents. Nor did they report what proportion of the 132 registered dogs were from what regions. Those may represent dogs from a more or less health conscious region or set of breeders and again not reflective of the population at large.

The response about relatedness and the addition on line 115 does not jibe with what the authors provided in their response to the reviewers. In the response the authors wrote “Unfortunately, we not able to perform pedigree analysis of the unregistered ECSs because we did not have reliable information about pedigrees; however, we can say that these animals were not directly related to each other. Instead, we were able to perform a pedigree analysis of the registered dogs, and it revealed an average inbreeding coefficient of 0.00941778.”

The authors say that they don’t have pedigree information on unregistered dogs yet Line 115 states “ In addition, pedigree visualization and analysis, not only of the registered dogs included in this study but also of three generations of each dog, was performed using Pedigraph v2.4 software [13]. Genealogical information of the dogs was obtained from a pedigree database (https://cockerspanieldatabase.info/en/).” They said they did not have the information on the unregistered dogs so how could they have evaluated the three generations of each dog?  Also, presumably they used the pedigraph program to calculate inbreeding coefficients? Did they do the inbreeding calculations only the 3 generation pedigrees of the registered dogs they collected? Reliability of inbreeding calculations for 3 generations is not viewed as reflective of relatedness. However, it very well may be that these dogs are unrelated at the grandparent level—a standard frequently used.  Furthermore they state they don’t have reliable pedigree information on the unregistered but state with conviction they are not related.

The authors added in the Czech data reference for English Cocker Spaniels in the introduction however the discussion of this data is unclear “Although there was a lack of peer review evaluating the allele frequency of prcd-PRA in ECS dogs, the allele frequency of prcd-PRA in this breed was 34% in the Czech Republic in 2011 [9], which is similar to that found in the same year by ECSCA.”

They did not address the reviewers’ concerns on males vs females in the registered population; only on the larger population that included unregistered dogs.  And the reader does not know the sex distribution among the registered dogs.

Linking the mutation to eye conditions yet being unable to accurately assess eyes does not create a cohesive research report for publication in a quality journal. In the eyes assessed the correspondence with the mutation is weak at best. The authors did not address the reviewers expressed concerns about this lack of congruence.